# Numerical Investigation of Heat Transfer Characteristics of scCO_2_ Flowing in a Vertically-Upward Tube with High Mass Flux

**DOI:** 10.3390/e24010079

**Published:** 2022-01-01

**Authors:** Kaigang Gong, Bingguo Zhu, Bin Peng, Jixiang He

**Affiliations:** School of Mechanical & Electronical Engineering, Lanzhou University of Technology, Lanzhou 730050, China; gkg2969562848@163.com (K.G.); pengb2000@163.com (B.P.); jxhejason@163.com (J.H.)

**Keywords:** supercritical carbon dioxide, heat transfer, high mass flux, pseudo-boiling

## Abstract

In this work, the heat transfer characteristics of supercritical pressure CO_2_ in vertical heating tube with 10 mm inner diameter under high mass flux were investigated by using an SST *k*-*ω* turbulent model. The influences of inlet temperature, heat flux, mass flux, buoyancy and flow acceleration on the heat transfer of supercritical pressure CO_2_ were discussed. Our results show that the buoyancy and flow acceleration effect based on single phase fluid assumption fail to explain the current simulation results. Here, supercritical pseudo-boiling theory is introduced to deal with heat transfer of scCO_2_. scCO_2_ is treated to have a heterogeneous structure consisting of vapor-like fluid and liquid-like fluid. A physical model of scCO_2_ heat transfer in vertical heating tube was established containing a gas-like layer near the wall and a liquid-like fluid layer. Detailed distribution of thermophysical properties and turbulence in radial direction show that scCO_2_ heat transfer is greatly affected by the thickness of gas-like film, thermal properties of gas-like film and turbulent kinetic energy in the near-wall region. Buoyancy parameters *Bu* < 10^−5^, *Bu** < 5.6 × 10^−7^ and flow acceleration parameter *K_v_* < 3 × 10^−6^ in this paper, which indicate that buoyancy effect and flow acceleration effect has no influence on heat transfer of scCO_2_ under high mass fluxes. This work successfully explains the heat transfer mechanism of supercritical fluid under high mass flux.

## 1. Introduction

As a basic physical phenomenon, the heat transfer process involves various application areas such as air conditioning refrigeration, heat pumps, thermal power plants, aerospace, electronic chip thermal management system and battery thermal management system. Convective heat transfer plays a significant role in above fields, high performance liquid convection not only enhances heat transfer efficiency but also improves system stability and safety [1,2]. Conventional convection intensification techniques fall into three categories, active method, passive method and compound method [3]. Active techniques include mechanical agitation of heat transfer fluid, vibration of heat transfer surface and vibration of heat transfer fluid. Passive techniques include artificial roughness (ribs and grooves), special shaped tubes, twisted tapes inserts, multiple swirl devices and longitudinal vortex generators.

As we all know, convectional heat transfer fluid such as oil, water, ethylene glycol and ethylene glycol mixture possess low thermal conductivity compared to solid metals, which makes these fluids have poor heat transfer performance as a heat transfer medium. Using nanofluid as heat transfer working medium is a new method to enhance heat transfer. Nanofluid, coined by Choi [4], refers to a fluid in which nanometer-sized particles with high thermal conductivity are suspended in conventional heat transfer basic fluids. Recent review articles shows that nanofluids significantly improve the heat transfer capability of conventional heat transfer fluids such as oil or water by suspending nanoparticles in these base liquids [5]. In addition to nanofluids, in recent years, liquid metals have rapidly emerged as some of the most attractive coolants for heat transfer enhancement duo to excellent thermophysical properties, including low melting point, high boiling point and high thermal conductivity. Deng et al. [2] systematically reviewed low-melting-point liquid metal convective heat transfer based on gallium-based and bismuth-based alloys from the perspectives of materials, mechanisms and applications. They draw a conclusion, compared with conventional convection technologies, as the most important advantages of liquid metal convective heat transfer are the ultra-high convective heat transfer coefficient, excellent stabilities at high temperatures, and efficient driving by a nonmechanical pump. MXene with a chemical formula of Ti_3_C_2_ is synthesized using wet chemistry method and suspended in pure olein palm oil to formulate a new class of heat transfer fluid was numerically investigate by Samylingam et al. [6]. They found that the heat transfer coefficient of MXene nanofluids increased by about 9% compared to Al_2_O_3_-water heat transfer fluid.

As a natural working fluid, carbon dioxide (CO_2_) has many advantages, such as environmental friendliness (ozone depleting potential, ODP = 0; effective global warming potential, EGWP = 0), non-toxic, chemically stable, non-combustible and low thermodynamic critical points (critical pressure: 7.38 Mpa, critical temperature: 31.04 °C). Because of the above excellent physical and chemical properties, supercritical CO_2_ (scCO_2_) is regarded as an attractive working fluid for the closed power cycles driven by various heat sources such as solar energy, nuclear energy and coal. In recent years, scCO_2_ Brayton cycle has drawn a lot of attention in the field of power generation [7,8]. Compared to supercritical water-power cycle, the scCO_2_ cycles have higher thermal efficiency [9]. In such a system, it is necessary for the heater to receive ultra-high heat flux heated by nuclear energy, solar energy or burning coal [10,11,12]. Considering a solar driven scCO_2_ power cycle, if the solar receiver is not well cooled by the scCO_2_ fluid, the solar receiver will be burned out. This phenomenon should be avoided in the design stage. In addition to that, due to the severe change of the thermophysical properties of scCO_2_ near the pseudo-critical temperature *T*_pc_ where the specific heat value reaches its maximum (see Figure 1), which results in the very complex heat transfer characteristics of scCO_2_. The data in Figure 1 is from the NIST standard database, Refprop 9.0. Therefore, studying the heat transfer characteristics and understanding the heat transfer mechanism of scCO_2_ is very important to improve system design and ensure system safety for scCO_2_ cycles.

Previous studies have shown that when the supercritical fluid is crossing the pseudo-critical temperature, there are three modes of heat transfer: normal heat transfer (NHT), heat transfer enhancement (HTE) and heat transfer deterioration (HTD). NHT refers to the wall temperature rise monotonously along the tube. HTE is the sudden increase of heat transfer coefficient near *T*_pc_. A sharp wall temperature rise is called HTD. When HTD occurs, it might lead to the heating surface failure because of burning out. Hence, great attention has been paid on HTD [13,14,15]. At present, the mechanism of HTD is mainly studied by numerical methods and experiments.

A series of experimental and numerical simulation studies on heat transfer characteristics of scCO_2_ flowing in heated mini-tubes were carried out Jiang’s group [16,17,18]. Jiang et al. [16,17] experimentally and numerically investigated the heat transfer of CO_2_ at supercritical pressures in a vertical mini tube with diameter of 0.27 mm at relatively low Reynolds numbers for upward and downward flows. They found that the buoyancy effect is weak when the heating is relatively strong, while the flow acceleration effect strongly influences the turbulence and reduces the heat transfer for high heat flux. Li et al. [18] experimentally studied the heat transfer of scCO_2_ flowing in vertical upward and downward flow with an inner diameter of 2.0 mm, they found that a sharp wall temperature rise occurred in vertical upward flow but was not found in downward flow.

Kim et al. [19] conducted an experimental study on heat transfer of scCO_2_ in vertical upward and downward flow with uniform heating, and, also, found the same results as Li et al. [18]. Kline et al. [20] experimentally studied the convective heat transfer to CO_2_ flowing in a vertical upward heated tubes at supercritical pressures, the onset of HTD in upward flows was identified for a wide range of conditions, and they found that HTD was observed within a range of inlet temperatures, outside of which the wall temperature behave monotonically increased. Some researchers have found that flow direction also has a great influence on heat transfer of scCO_2_. They found that a sharp wall temperature rise occurred in vertical upward flow but was not found in downward flow. Zhang et al. [21] performed an experiment to study heat transfer of scCO_2_ in a 16 mm diameter tube with low mass flux. A special heat transfer behavior was observed, heat transfer at lower mass flux was not deteriorated but rather enhanced with a rising heat transfer coefficient. They suggested that this phenomenon was possibly induced by the combined effects of strong buoyancy effect and high specific heat of the fluid. As a continuity of their experiments, Zhang et al. [22] performed numerical works to explore the mechanisms behind different heat transfer behaviors of scCO_2_. The V2F model is recommended to predict heat transfer behaviors of sCO_2_ at low mass flux. For normal mass flux cases, the SST *k-ω* model can well capture the heat transfer deterioration and recovery. Fan et al. [23] numerically investigated the heat transfer of scCO_2_ inside the upward circular tube near the pseudo-critical region with non-uniform heating vertical tubes, they found that HTD is more likely to be induced by local thickening of the viscous sublayer. Meanwhile, when heat flux and mass flux are both high, the existing buoyancy criterions are no longer applicable.

Our literature survey showed that supercritical fluid has been regarded as homogeneous structure. Based on single-phase assumption of supercritical fluid, a large number of studies have attributed the mechanism of HTD in supercritical fluids to buoyancy and flow acceleration effect. However, the inconsistent results or even opposite results could be found in the existing literature [24,25]. The criterion number of the onset of buoyancy and flow acceleration effect is only suitable for their own experimental conditions and not widely applicable. Hence, using single-phase assumption to analyze supercritical heat transfer is facing great challenges. In recent years, the cognition of single-phase of supercritical fluid is questioned by physicists. Recent studies have shown that the supercritical fluid has a heterogeneous structure, a Widom line (WL) separates the supercritical fluid into a liquid-like region and a gas-like region (see Figure 2) [26,27,28].

Banuti [29] proved the existence of supercritical pseudo-boiling when crossing the Widom-line, he argued that the supercritical pseudo-boiling does not occur in a phase equilibrium but takes place over a finite temperature interval (*T*^−^ and *T*^+^), where *T*^−^ is less than *T*_pc_ and *T*^+^ is greater than *T*_pc_. In other words, the bulk fluid temperature below *T*^−^ is a liquid-like fluid, beyond *T*^+^ is a gas-like fluid. The pseudo-boiling enthalpy Δ*i* at supercritical pressure consists of latent heat Δ*i*_lat_ and sensible heat Δ*i*_sen_, the former is used to overcome intermolecular attraction for phase transition, and the latter is used to raise the temperature. Recent experimental studies and molecular dynamics simulations confirm Banuti’s theoretical analysis [30,31].

Thus, the objective of this work is to numerically study the heat transfer behaviors of scCO_2_ under high mass flux, to discuss the effects of various parameters on heat transfer, and to understand the mechanisms based on pseudo-boiling theory. A physical model of scCO_2_ heat transfer in vertical heating tube was established containing a gas-like layer near the wall and a liquid-like fluid layer, the two regions are interfaced at *T* = *T*^+^. Detailed distribution of thermophysical properties and turbulence in radial direction showed that supercritical heat transfer is greatly affected by the thickness of gas-like film, thermal properties of gas-like film and turbulent kinetic energy in the near-wall region. The low thermal conductivity of the gas-like layer resulting in large thermal resistance and the reduction of turbulent kinetic energy near the wall are the main reasons for the weakening of heat transfer and the occurrence of sharp rise of the wall temperature or heat transfer deterioration (HTD). Our work can lead to a better understanding of the mechanisms of supercritical heat transfer.

## 2. Numerical Methods

### 2.1. Physical Model

The simulating physical model of vertically upward circular tube with 10 mm inner diameter in present study is established as shown in Figure 3. The calculation domain consists of inlet adiabatic section, heating section and outlet adiabatic section. The length of the inlet adiabatic section is 500 mm to ensure the CO_2_ at the inlet of the heating section fully developed. The effective heating length of the tube is 2000 mm, which is the main section of studying the heat transfer characteristics of scCO_2_. The outlet adiabatic section (300 mm) is provided to prevent reverse flow phenomenon. The flow direction is along the positive direction of z-axis. The direction of gravity is opposite to the flow direction.

### 2.2. Governing Equations and Solution Procedure

For the vertical upward uniform heating tube, considering the symmetry of heat transfer and flow, in order to save calculation cost and reduce calculation time, we assume that the fluid flow is two-dimensional flow. For steady-state flow and heat transfer, the governing equation includes conservation of mass, momentum, and energy. The Reynolds-averaged governing equations were employed in this study, and above equations described in Cartesian coordinate system are as follows:

Continuity equation:(1)$$\frac{\partial \left(\rho u_{i}\right)}{\partial x_{i}}=0$$

Momentum equation:(2)$$\frac{\partial \left(\rho u_{i}u_{j}\right)}{\partial x_{j}}=-\frac{\partial p}{\partial x_{i}}+\frac{\partial }{\partial x_{j}}\left(\mu \frac{\partial u_{i}}{\partial x_{j}}-\bar{\rho u_{i}^{\prime }u_{j}^{\prime }}\right)+\rho g_{i}$$

Energy equation:(3)$$\frac{\partial \left(\rho u_{i}i\right)}{\partial x_{i}}=\frac{\partial }{\partial x_{i}}\left[\mu \left(\frac{1}{\mathrm{Pr}}+\frac{\mu_{t}/\mu }{{\mathrm{Pr}}_{t}}\right)\frac{\partial i}{\partial x_{i}}\right]$$

The selection of a suitable turbulence model is of great importance for the accuracy of numerical calculation results of supercritical fluid flow and heat transfer. Recent review articles [32] have shown that the SST *k*-*ω* turbulence model can obtain more accurate calculation results than other turbulence models. Therefore, the SST *k*-*ω* turbulence model is used for present numerical calculation The SST *k-ω* turbulence model is briefly described as follows:

The transport equation *k*:(4)$$\frac{\partial \left(\rho u_{i}k\right)}{\partial x_{i}}=\frac{\partial }{\partial x_{j}}\left[\left(\mu +\frac{\mu_{t}}{\sigma_{k}}\right)\frac{\partial k}{\partial x_{j}}\right]+G_{k}-Y_{k}$$

The transport equation *ω*:(5)$$\frac{\partial \left(\rho u_{i}\omega \right)}{\partial x_{i}}=\frac{\partial }{\partial x_{j}}\left[\left(\mu +\frac{\mu_{t}}{\sigma_{\omega }}\right)\frac{\partial \omega }{\partial x_{j}}\right]+G_{\omega }-Y_{\omega }+D_{\omega }$$

In Equations (1)–(5), when subscripts *i* and *j* are 1 and 2, they, respectively, represent two different directions of *y* and *z*. *u* is the velocity vector, *ρ* is density, *μ* is viscosity coefficient, *i* is specific enthalpy, the acceleration of gravity g is equal to 9.8 m/s^2^, *μ*_t_ is the coefficient of turbulent viscosity, *Pr_t_* is turbulent Prandtl number, *G**_k_* is the turbulence production term, *Y**_k_* represents the dissipation term of turbulent kinetic energy *k* due to turbulence, *G**_ω_* represents the generation term of specific dissipation rate, *Y**_ω_* is the dissipation term of specific dissipation rate, *D**_ω_* is the cross diffusion term and other constant terms and function terms are detailed in reference [33].

Numerical calculations were carried out using ANSYS FLUENT 15.0. The inlet and the outlet condition are set as the mass-flow-inlet and the pressure-outlet, respectively. The heating section is set as the constant wall heat flux, and the adiabatic section is set as the adiabatic boundary, and the fluid-wall is set as no slip condition. The finite volume method was used to discretize the governing equation. In the discrete scheme, the second-order upwind algorithm is adopted and the SIMPLEC algorithm is employed to solve the pressure–velocity coupling equation. The real gas model in Fluent software is called to ensure the demand for physical characteristic of sCO_2_ during the numerical simulation. When the relative residual of continuity equation is less than 10^−5^ and the relative residual of other governing equation is less than 10^−7^, the numerical calculation is considered to be converged. Detailed parameters are listed in Table 1. The solving process of the governing equation is shown in Figure 4.

The bulk fluid temperature at a cross-section is defined as:(6)$$T_{b}=\frac{\int \rho uc_{p}TdA}{\int \rho uc_{p}dA}$$
where *d**A* is an elemental area of the tube across-section.

The bulk fluid enthalpy at a cross-section is defined as:(7)$$i_{b}=\frac{\int \rho uidA}{\int \rho udA}$$

The local heat transfer coefficient *h* can be expressed as:(8)$$h=\frac{q_{w}}{T_{w}\;-\;T_{b}}$$
where *T*_w_ is the inner wall temperature and *q*_w_ is the inner wall heat flux.

### 2.3. Grid Independence Verification and Model Validation

The computational domain grid is generated by ANSYS ICEM software and adopts a structured grid. Rectangular grid had been used in present numerical calculations. Firstly, the distance between the inner wall and the first node is set as 0.001 mm, and the growth ratio is 1.1. On this basis, due to the dramatic physical property change of scCO_2_, the grid near the wall is locally densified. Secondly, in order to test the independence of computational grid, five grid systems type are employed to test the independence of mesh as shown in Figure 5. As can be seen from Figure 5, when the total number of nodes in the grid is greater than 85,000, the calculation results have no significant deviation with the increase of the total number of nodes. Therefore, the total number of nodes is finally determined to be 85,000 in current study. Table 2 lists the setting value of operating parameters.

In order to verify the reliability and accuracy of the calculation method and turbulence model in this paper, the numerical results are compared with the experimental results of Ref. [34], at the same time the inner diameter of the vertical tube is 10 mm. Figure 6 compares the calculated results with the experimental data. Since there is not turbulence model for supercritical fluids, so the numerical simulation results may deviate from the experimental data in some working conditions. Figure 6 shows that the numerical results using the SST k-ω model have the well agreement with the experimental results qualitatively. Hence, the numerical model of the SST *k*-*ω* model used in this paper is feasible and reliable. The cases for present computations are listed in Table 3.

## 3. Results and Discussion

### 3.1. Effect of Inlet Temperature on scCO_2_ Heat Transfer

Figure 7 demonstrates the effects of inlet temperature on the temperatures of inner wall. The abscissa is the bulk fluid enthalpy *i*_b_, and the ordinate is the inner wall temperature and the pseudo-critical enthalpy *i*_pc_ is also marked.

Figure 7 shows that with the same pressure, heat flux and mass flux, there is a prominent temperature peak was observed when inlet temperature is less than pseudo-critical temperature, which indicates heat transfer deterioration (HTD) occurs. We further observed that the wall temperature peak point moves towards the inlet with the decrease of T_in_. However, the wall temperature increases monotonously when inlet temperature is greater than pseudo-critical temperature. From the observations described above, it can be seen that the heat transfer characteristics at supercritical pressure are significantly affected by the inlet temperature, HTD is closely related to inlet temperature. Here, the reason for the occurrence of HTD requires that the inlet temperature is less than the pseudo-critical temperature is discussed based on the pseudo-boiling concept. Figure 8 shows a physical model of scCO_2_ heat transfer in vertical heating tube, and the inlet temperature is less than *T*_pc_.

According to the pseudo-boiling theory, supercritical phase boiling takes place over a finite temperature interval *T*^−^ and *T*^+^, thus, an interface thickness s to supercritical pseudo-boiling exists (see Figure 8). When “subcooled” fluid at supercritical pressure flows into a vertical heated tube, a gas-like layer near the wall is formed when *T*_w_ > *T*_pc_ > *T*_b_, and the tube core is liquid-like fluid (see Figure 8), the two regions are interfaced at *T* = *T*
^+^, *T*^−^ and *T*^+^ is determined in Ref. [35]. Due to the low thermal conductivity of the gas-like phase, the thick gas-like layer results in large thermal resistance, which will severely inhibit heat diffusion from the tube wall to the core flow and, finally, causes sharp rise of wall temperature or HTD, which is similar to film boiling at subcritical pressure. However, when the inlet temperature is higher than the pseudo-critical temperature, the bulk fluid is pure gas-like fluid, the heat transfer characteristic accords with the single-phase convective heat transfer, thus, the wall temperature rises monotonously along the bulk fluid enthalpy.

The *T*^−^ and *T*^+^ are determined by an approach based on intersections of specific enthalpy asymptotes, which is also originally proposed by Banuti [29]. The method determined *T*^−^ and *T*^+^ used in this work is illustrated in Figure 9 for CO_2_ at 8.5 MPa, it can be well explained. The enthalpy-temperature curve at subcritical pressure of 6 MPa also is presented, it can be seen that phase transition takes place at a constant saturation temperature *T*_sat_ at subcritical. Unlike subcritical pressure, supercritical phase transition occurs over a finite temperature interval *T^−^* and *T*^+^ (see black line). The blue and green curves are the liquid-like and vapor-like enthalpy asymptotes, respectively. In this paper, the asymptotes function of two lines are represented by [35]:(9)$$i_{\mathrm{LL},\mathrm{asymptotes}}=c_{p,L}\left(T-T_{L}\right)+i_{0,L}$$
(10)$$i_{\mathrm{VL},\mathrm{asymptotes}}=c_{p,V}\left(T-T_{V}\right)+i_{0,V}$$
where $c_{p,L}=c_{p}\left(P=P_{c},T=0.75T_{c}\right)$, $i_{0,L}=i\left(P=P_{c},T=0.75T_{c}\right)$, $c_{p,V}=c_{p}\left(P=0,T=T_{c}\right)$, $i_{0,V}=i\left(P=0,T=T_{c}\right)$, $T_{L}=0.75T_{c}$, $T_{L}=T_{c}$, *P*_c_ is the critical pressure and *T*_c_ is the critical temperature.

The red line is the pseudo-critical asymptotes, which is tangent to the enthalpy line of 8 MPa at the pseudo-critical point, and the asymptotes’ function can be expressed as:(11)$$i_{\mathrm{pc},\mathrm{asymptotes}}=c_{p,\mathrm{pc}}\left(T-T_{\mathrm{pc}}\right)+i_{\mathrm{pc}}$$
where *c*_p,pc_ and *i*_pc_ are the specific heat capacity and enthalpy at pseudo-critical point, respectively.

Then, the liquid-like enthalpy asymptotes and tangent line have a crossing point *A* to determine *T*^−^, and the vapor-like enthalpy asymptotes and tangent line have a crossing point *B* to determine *T*^+^ (see Figure 9).

### 3.2. Effect of Heat Flux on scCO_2_ Heat Transfer

Figure 10 demonstrates the effects of heat fluxes on inner wall temperatures. The abscissa is the bulk fluid enthalpy *i*_b_ and the ordinate is the inner wall temperature, and pseudo-critical enthalpy *i*_pc_ is marked. When the inlet temperature, pressure and mass fluxes keep constant, with increase of inner wall heat fluxes *q*_w_, wall temperatures increase. When the *q*_w_ is 200 kW/m^2^, with increase of bulk fluid enthalpy, the wall temperature rises smoothly, this belongs to NHT.

When the *q*_w_ is increased to 350 kW/m^2^, there is a temperature peak appearing ahead of the pseudo-critical point, which belongs to HTD mode. In case of higher heat flux (400 kW/m^2^), the wall temperature had a noticeable peak, and the peak point moves towards the lower bulk fluid enthalpy region. We also observed that for the high heat flux the wall temperature reach a constant value after the peak, this is because, when the bulk fluid temperature is greater than the pseudo-critical temperature, the bulk fluid is pure gas-like fluid, and its heat transfer mechanism is single-phase convective heat transfer, so the wall temperature shows a smooth increase. In order to analyze the reasons of different heat fluxes leading to different heat transfer behavior, the characteristic cross section D, E and D′, E′ are marked in Figure 10 for the convenience of discussion.

The detailed distributions of thermophysical properties and turbulence in radial direction at above characteristic cross section D, E and D′, E′ are presented in Figure 11. In each panel, the radial position of *T*^+^ is also marked, the gas-like film thickness *δ* is defined as the distance from the radial position of *T*^+^ to the tube inner wall. As can be seen from the Figure 11, the thickness of gas-like film at characteristic cross section D is greater than that at characteristic cross section E′, the thickness of gas-like film at characteristic cross section E is much larger than that at characteristic cross E′, that is to say, the thickness of gas-like film increases with the increase of heat flux. The thermal conductivity of gas-like layer generally is very low (see Figure 11a), close to the thermal conductivity of air, which results in a large thermal resistance, thus, the heat transfer is impaired. Figure 11b shows the distributions of specific heat at different characteristic cross section, the specific heat at characteristic cross sections D and E in the near wall region is less than that at characteristic cross sections D′ and E′ (see Figure 11b), indicating the heat absorption capacity of the gas-like film is also weak, which is also detrimental to heat transfer. Figure 11c shows the radial temperature distribution, the temperature near the wall is the highest, and the temperature gradient in the near wall region is larger than that in the core flow region. In addition to the above reasons, we also found that the reduction of turbulent kinetic energy near the wall would also deteriorate heat transfer (see Figure 11d). In summary, it is concluded that supercritical heat transfer is greatly affected by the thickness of gas-like film, thermal properties of gas-like film and turbulent kinetic energy in the near-wall region.

### 3.3. Effect of Mass Flux on scCO_2_ Heat Transfer

The effect of mass flux on heat transfer of scCO_2_ was investigated at *P* = 8 MPa, *q*_w_ = 400 kW/m^2^, *G* = 1500–2500 kg/m^2^s. The distribution curves of wall temperature with bulk fluid enthalpy are demonstrated in Figure 12. Figure 12 shows that the wall temperatures are significantly influenced by the mass flux under heating condition, the wall temperature greatly decreases as the mass flux increases. The wall temperature has a remarkable peak at *G* = 1500 kg/m^2^s, when the wall temperature increases monotonously at *G* = 2500 kg/m^2^s, indicating the higher mass flux can remove HTD and improve heat transfer.

The reasons for higher mass flux can eliminate HTD and improve heat transfer will be explained. The working conditions of mass flux *G* = 1500 kg/m^2^s and *G* = 2500 kg/m^2^s were selected for analysis. For the convenience of discussion, the characteristic cross section F, G and F′, G′ were selected to analysis the different heat transfer behavior resulting from different mass fluxes, as shown in Figure 12. Figure 13 illustrates the detailed distributions of thermophysical properties and turbulence in radial direction at above characteristic cross section F, G and F′, G′. In each panel, the gas-like film thickness *δ* is also marked. It can be seen from Figure 13, in general, the thickness of gas-like film decreases with the increase of mass flux. The thickness of gas-like film at the cross-section G corresponding to the wall temperature peak is the largest. Because of the low thermal conductivity of the gas-like film and the low heat absorption capacity of the gas-like film near the wall (see Figure 11a,b), therefore, heat transfer is weakened at the characteristic section G. Figure 13c shows the distribution of density in radial direction. We can see that the near-wall zone is occupied by gas-like fluid, and the liquid-like fluid is in the core flow region. Except for the thickness and thermal properties of gas-like film, the turbulent kinetic energy near the wall, also, has a great influence on heat transfer. The turbulent kinetic energy near the wall is much higher at high mass flux than at low mass flux, as shown in Figure 13d. Thinner gas-like film and greater turbulent kinetic energy near the wall are the main reasons for heat transfer enhancement with increasing mass flux.

### 3.4. Buoyancy Effects

Figure 14 illustrates schematically the buoyancy phenomena with fluid flow upward in a vertical heated tube at supercritical pressures. Buoyancy effect refers to when the supercritical fluid flows into the heating tube, due to its severe variable physical properties near *T*_pc_, the radial density difference leads to the formation of strong buoyancy *F*_b_, the buoyancy *F*_b_ changes the velocity distribution of the fluid near the wall and, then, affects the shear force *τ* and turbulent kinetic energy, eventually led to the abnormal phenomena of heat transfer.

Jackson et al. [36] put forward a criterion number *Bu* of the onset of buoyancy effect for scCO_2_ in vertical pipe. The *Bu* number can be expressed as:(12)Bu=GrReb2.7
(13)$$Gr=\frac{g\rho_{b}(\rho_{b}-\rho_{\mathrm{ave}})d_{in}^{3}}{\mu_{b}^{2}},\;Re_{b}=\frac{Gd_{in}}{\mu_{b}},\;\rho_{\mathrm{ave}}=\frac{\int_{T_{b}}^{T_{w}}\rho dT}{T_{w}-T_{b}}$$

Another criterion number, *Bu** functions to characterize the buoyancy effect in vertical pipe is expressed as [34]:(14)Bu*=Gr*Reb3.425Prb0.8
(15)$$Gr^{*}=\frac{g\beta_{b}d_{in}^{4}q_{{}_{w}}}{\upsilon_{b}^{2}\lambda_{b}},Re_{b}=\frac{Gd_{in}}{\mu_{b}},\;Pr_{b}=\frac{\mu_{b}c_{p}}{\lambda_{b}}$$

Jackson et al. [36] suggested that when *Bu* > 10^−5^, buoyancy has of significance effect on supercritical fluid heat transfer, However, when *Bu* < 10^−^^5^, the influence of buoyancy on heat transfer can be ignored. Jackson and Hall [37] believed that the buoyancy effect is weak when *Bu** < 5.6 × 10^−^^7^.

The effect of buoyancy on heat transfer of scCO_2_ was investigated at *P* = 8 MPa, *q*_w_ = 400 kW/m^2^ and *G* = 1500–2500 kg/m^2^s. Figure 15 shows the variations of *Bu* and *Bu** under different mass flux. As mass flux increases from 1500 to 2500 kg/m^2^s, *Bu* and *Bu** clearly decrease. This indicates that the effect of buoyancy is weaker with the increase of mass flux. More importantly, at three different mass fluxes, *Bu* are all less than 10^−5^, and *Bu** are all less than 5.6 × 10^−7^ in the whole enthalpy region (see Figure 15). The above results show that buoyancy effect has no effect on heat transfer when mass flux is higher.

### 3.5. Flow Acceleration Effect

Mc Eligot [38] proposed the *K*_v_ number to characterize the flow acceleration effect on supercritical heat transfer:(16)$$K_{v}=\frac{4q_{w}d_{\mathrm{in}}\beta_{b}}{Re_{b}^{2}\mu_{b}c_{p}}$$

Mc Eligot [38] held that when *K*_v_ is less than 3 × 10^−6^, the flow acceleration effect is not considered.

The effect of flow acceleration on heat transfer of scCO_2_ also was studied at *P* = 8 MPa, *q*_w_ = 400 kW/m^2^ and *G* = 1500–2500 kg/m^2^s. Figure 16 reflects the local *K*_v_ distributions with bulk fluid enthalpy under different mass fluxes. In general, *K*_v_ decreases with the increase of mass flux, and *K*_v_ are all less than 3 × 10^−^^6^ in the whole enthalpy region for three different mass fluxes, indicating flow acceleration effect also has no influence on heat transfer of scCO_2_ under high mass flux.

## 4. Conclusions

The heat transfer behaviors of supercritical pressure CO_2_ in vertical heating tube under high mass flux were investigated using the SST *k-ω* turbulent model, the influences of inlet temperature, heat flux, mass flux, buoyancy and flow acceleration on the heat transfer of supercritical pressure CO_2_ were discussed. The main results are summarized as follows:The occurrence of HTD is closely related to inlet temperature. When inlet temperature is less than pseudo-critical temperature, a prominent temperature peak was observed ahead of the pseudo-critical point. However, the wall temperature increases monotonously when inlet temperature is greater than pseudo-critical temperature. The wall temperature increases as the heat flux increases, and the HTD occurs at high heat flux. Increasing mass flux can enhance heat transfer and even eliminate heat transfer deterioration. At *P* = 8 Mpa and *G* = 1500 kg/m^2^s, when the heat fluxes doubled, the average wall temperature increased by 3.82 times, however, when the mass flux increases 1.6 times, the average wall temperature decreases 2.57 times.At high mass flux, our results show that the buoyancy criterion number *Bu* < 10^−5^ and *Bu** < 5.6 × 10^−^^7^, they are both all less than their critical value in the whole enthalpy region, and flow acceleration criterion number *K*_v_ also is less than their critical value 3.6 × 10^−6^ in the whole enthalpy region, indicating the influences of buoyancy and flow acceleration on supercritical heat transfer can be neglected. Thus, the mechanism of supercritical heat transfer at high mass flux cannot be explained by buoyancy and flow acceleration effect.Supercritical pseudo-boiling is used to do with the heat transfer of scCO_2_. A physical model of scCO_2_ heat transfer in vertical heating tube was established containing a gas-like layer near the wall and a liquid-like fluid layer, the two domains are interfaced at *T* = *T*^+^. Due to the low thermal conductivity of the gas-like phase, the thick gas-like layer results in large thermal resistance, and, finally, causes sharp rise of wall temperature or HTD. Further analysis found supercritical heat transfer is extremely affected by the thickness of gas-like film, thermal properties of gas-like film and turbulent kinetic energy in the near-wall region.

## Figures and Tables

**Figure 1 entropy-24-00079-f001:**
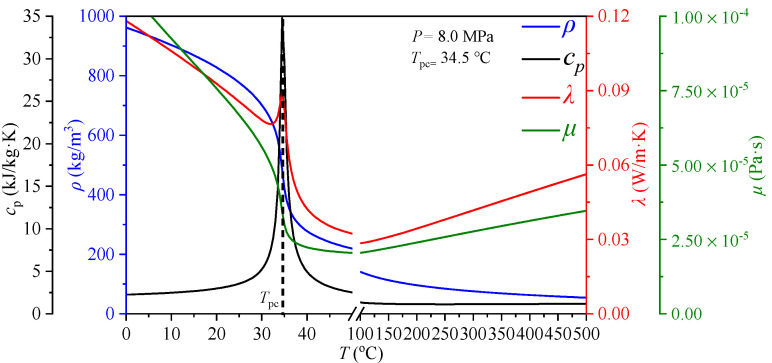
Thermal property variations versus temperature of CO_2_ at *P* = 8 Mpa.

**Figure 2 entropy-24-00079-f002:**
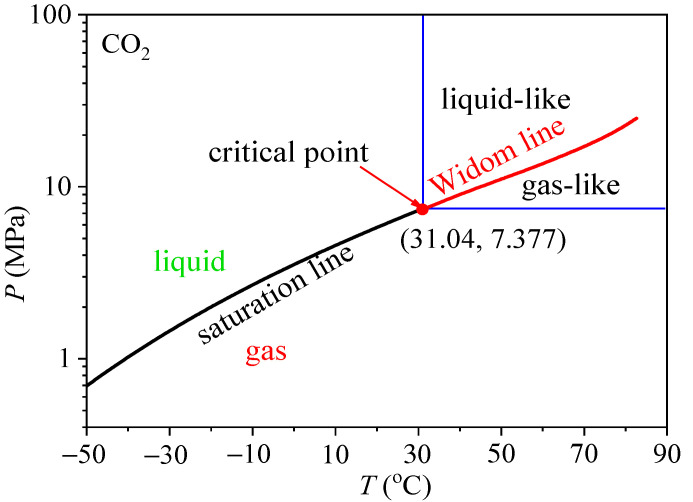
*P*-*T* phase diagram of CO_2_ at subcritical pressure and supercritical pressure.

**Figure 3 entropy-24-00079-f003:**
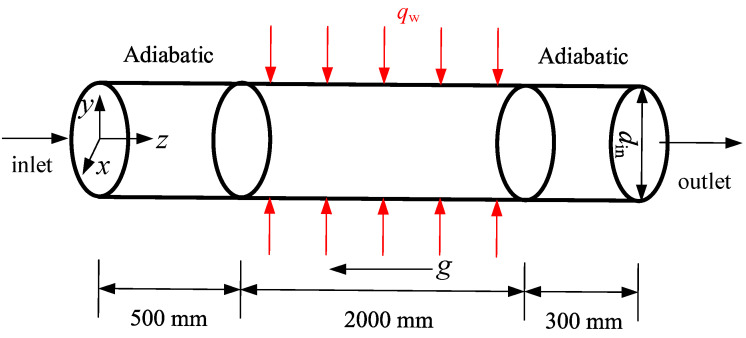
The physical model of vertically upward tube in present study.

**Figure 4 entropy-24-00079-f004:**
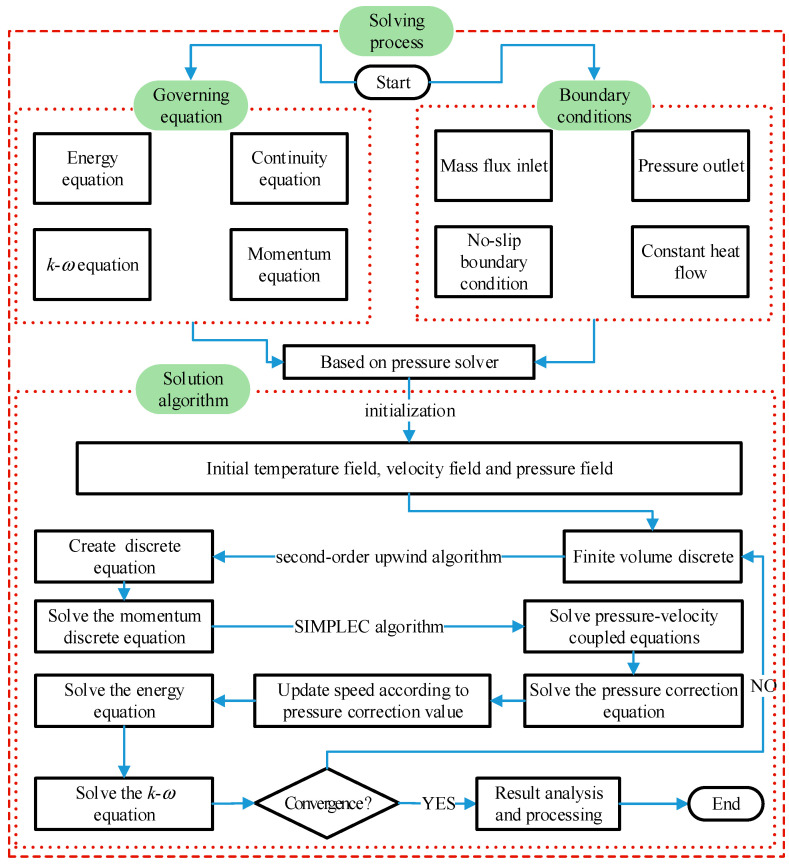
Flow chart of numerical solutions.

**Figure 5 entropy-24-00079-f005:**
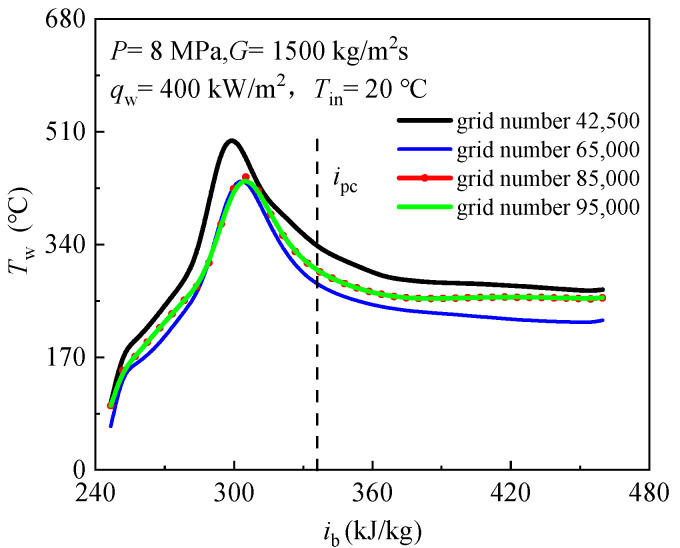
Mesh independence verification.

**Figure 6 entropy-24-00079-f006:**
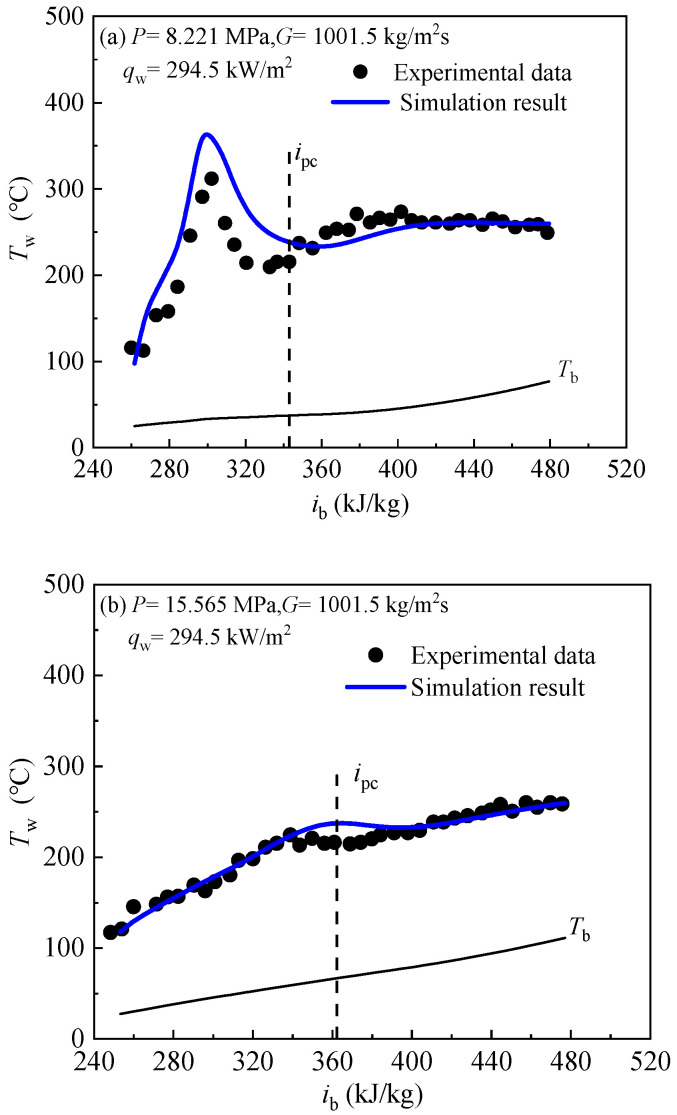
The comparison between present simulation and experimental data [34]. (**a**) Deteriorating heat transfer. (**b**) Normal heat transfer.

**Figure 7 entropy-24-00079-f007:**
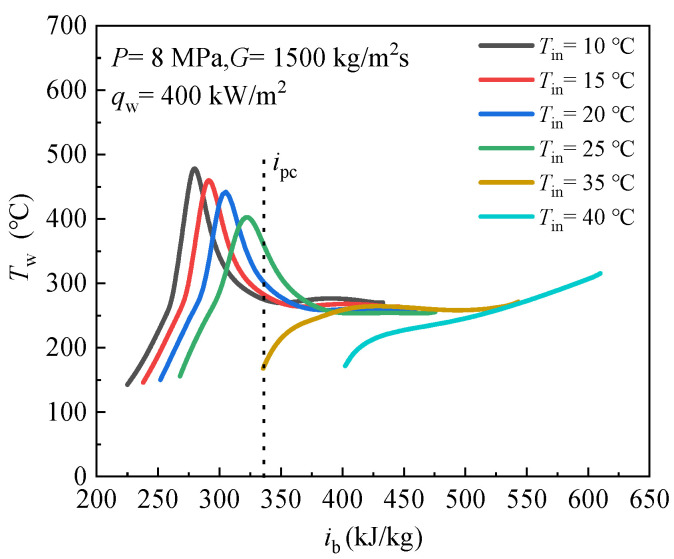
Effect of inlet temperature on CO_2_ heat transfer at supercritical pressure.

**Figure 8 entropy-24-00079-f008:**
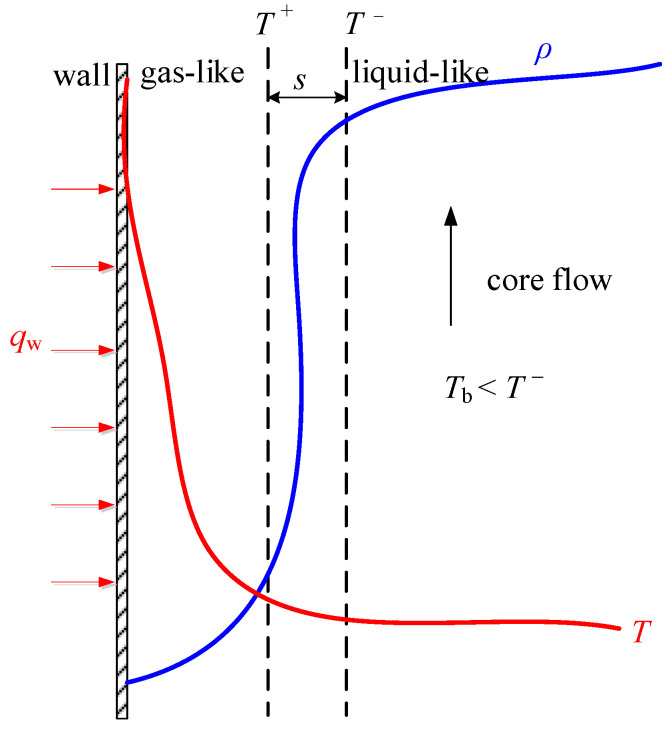
Physical model of scCO_2_ heat transfer in vertical heating tube.

**Figure 9 entropy-24-00079-f009:**
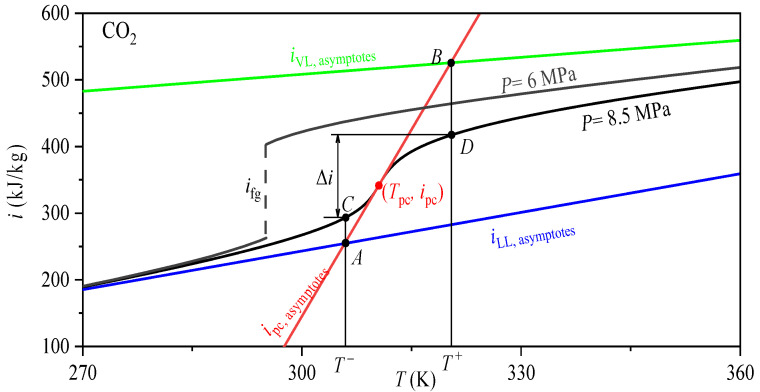
The schematic diagram for determining the *T*^−^ and *T*^+.^

**Figure 10 entropy-24-00079-f010:**
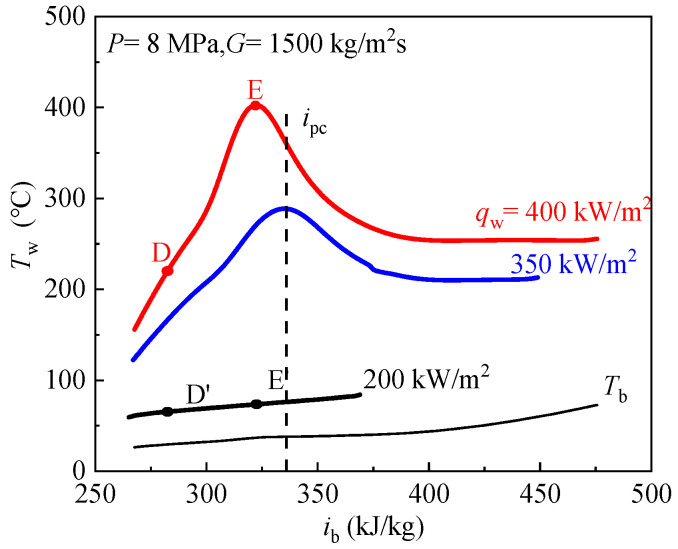
Effect of heat flux on CO_2_ heat transfer at supercritical pressure.

**Figure 11 entropy-24-00079-f011:**
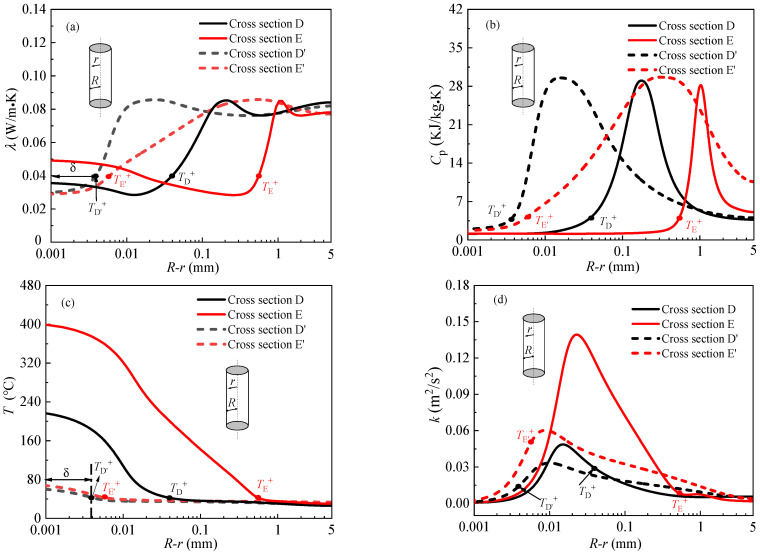
The distributions of thermophysical properties and turbulence in radial direction. (**a**) Thermal conductivity. (**b**) Specific heat. (**c**) Temperature. (**d**) Turbulent kinetic energy

**Figure 12 entropy-24-00079-f012:**
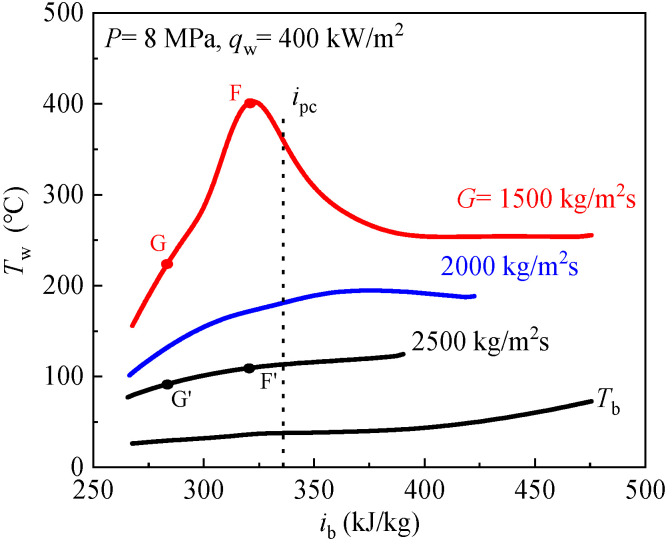
Effect of mass flux on CO_2_ heat transfer at supercritical pressure.

**Figure 13 entropy-24-00079-f013:**
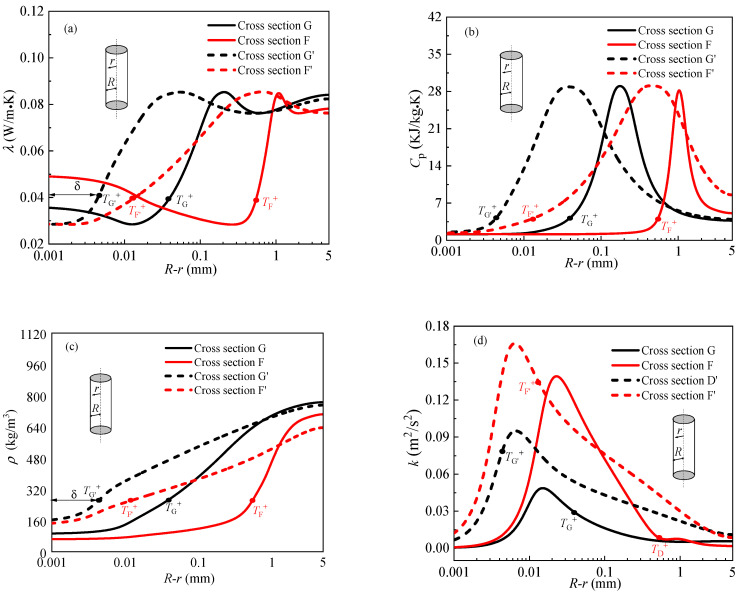
The distributions of thermophysical properties and turbulence in radial direction. (**a**) Thermal conductivity. (**b**) Specific heat. (**c**) Density. (**d**) Turbulent kinetic energy

**Figure 14 entropy-24-00079-f014:**
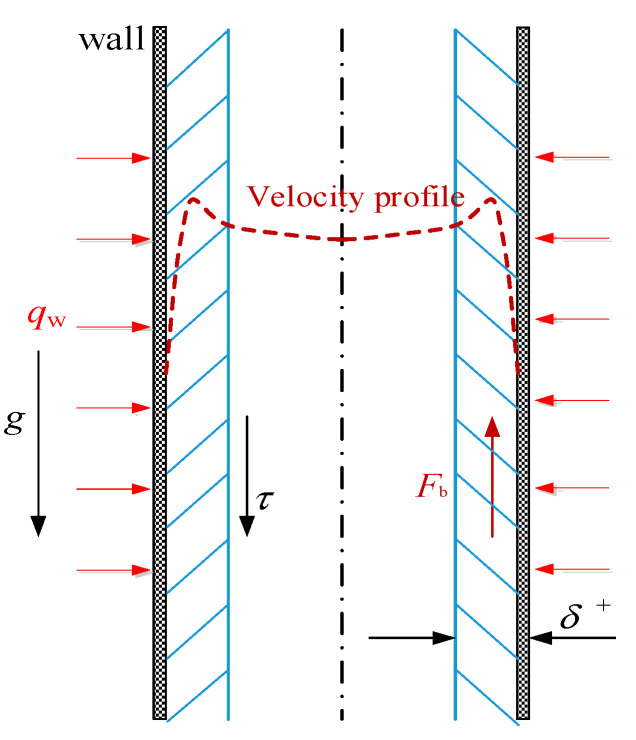
Schematic diagram of the buoyancy effects.

**Figure 15 entropy-24-00079-f015:**
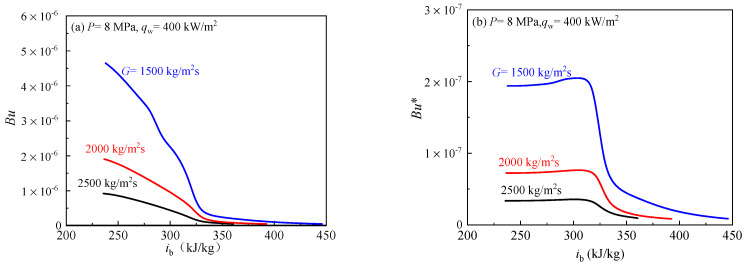
The variations of buoyancy at different mass fluxes. (**a**) The distribution curve of criterion number *Bu* along the bulk fluid enthalpy. (**b**) The distribution curve of criterion number *Bu** along the bulk fluid enthalpy.

**Figure 16 entropy-24-00079-f016:**
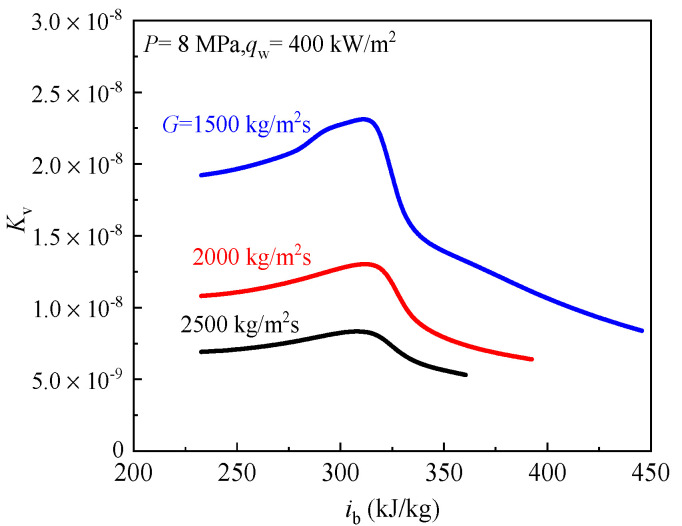
The distributions of *K*_v_ under different mass flux.

**Table 1 entropy-24-00079-t001:** Relative residual.

Residual	Continuity	x-Velocity	y-Velocity	z-Velocity	Energy	*k*	*ω*
Value	10^−5^	10^−7^	10^−7^	10^−7^	10^−7^	10^−7^	10^−7^

**Table 2 entropy-24-00079-t002:** Settings value of model parameter.

Under-Relaxation Factors	Pressure	Density	Volume Force	Momentum	Turbulent Kinetic Energy	Specific Dissipation Rate	Turbulent Viscosity	Energy
Value	0.3	1	1	0.7	0.8	0.8	0.85	0.8

**Table 3 entropy-24-00079-t003:** The cases for present computations.

Cases	*P* (MPa)	*G* (kg/m^2^s)	*q*_w_ (kW/m^2^)	*T*_b,in_ (°C)
heating	8	1500	200–400	10/15/20/25/35/40/50
		2000	200–400	20
		2500	200–400	20

## Data Availability

Not applicable.

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
