# Peer review of "Numerical Investigation of Heat Transfer Characteristics of scCO2 Flowing in a Vertically-Upward Tube with High Mass Flux"

_entropy, 2022, doi:10.3390/e24010079_

Round 1

Reviewer 1 Report

Comments on “Numerical investigation of heat transfer characteristics of scCO2 flowing in a vertically-upward tube with high mass flux”

  • Check the English and make sure that all typos and grammatical errors are addressed in the next version of the paper.
  • Introduction could be improved by reviewing state-of-the-art studies on heat transfer characteristics of advanced thermal engineering fluids (nano-suspensions, liquid metals, nano-oils,…) and then narrow it down to SC-CO2 as potential HTF. Searching the literature, following papers are suggested to be read and used: Two-phase frictional pressure drop with pure refrigerants in vertical mini/micro-channels. Effects of magnetic field on micro cross jet injection of dispersed nanoparticles in a microchannel.  Heat transfer improvement in a double backward-facing expanding channel using different working fluids
  • Add a table showing the range of operating parameters considered in the modelling.
  • Add an algorithm of solution showing which equations and on which order they are being solved.
  • What is the physical concept of BU number in the paper? It must be elaborated.
  • Add more quantitative data to the abstract and also the conclusion sections.

All in all, the paper can be accepted once above comments are addressed.

Reviewer 2 Report

Review

Kaigang Gong, Bingguo Zhu, Bin Peng, Jixiang He:

Numerical investigation of heat transfer characteristics of scCO2 flowing in a vertically upward tube with high mass flux.

In the article authors show results obtained by numerical modelling of a heated pipe. They find effects connected to the inhomogeneity of the thermophysical properties around the pseudocritical point.

All in all the work is well performed and the analysis is adding to the understanding of the effects.  

General remarks:

  • The use of English language is not well. Some sentences remain unclear. It is highly recommended to rework the whole article preferably by a native speaker.
  • Throughout the article formal mistakes are abundant. It is indispensable that authors go through the finished manuscript and look for such details.
  • Authors introduce new methods in the results and discussion chapter ( namely Jackson’s buoyancy and McEliggot’s flow acceleration. The relevant paragraphs should be moved.
  • It is strictly necessary to show how T+ is calculated. That is the reason for “major revisions”

Specific comments:

  • Line 16 authors should omit “the two regions are interfaced at T=T+” this is far too specific for an abstract.
  • Line 24-25 The text is left over from the template. This is one example, where a thorough review by the authors would enhance the overall impression considerably.
  • Line 28: The authors might want to say “non-combustible”?
  • Line 33/34 another example of problematic wording. Authors probably want to say “…burned out if not well cooled by the cooling fluid”
  • Figure 1: The reviewer is unable to find the line for r
  • Figure 1: If the data are adapted from sources [7,8] it should be indicated.
  • Figure 2: WL means Widom Line adding “line” is superfluous.
  • Figure 3: what does “circular” mean in the description?
  • line 127 the sentence is not to be understood
  • line 129: there is “flow profile” missing
  • line 150-152 reformulate sentence.
  • Paragraph line 160 to 164.: first sentence is unclear. “u” is not in Nomenclature, same with µT. ALL (s, G, Y, D) symbols are to be explained. You can’t refer to a literature source!
  • Line 165: probably authors meant to say: “Numerical calculations were carried out (better: performed) using ANSYS FLUENT 15.0”?
  • Line 171-173 sentence unclear: Did authors use data from webbook.nist.gov? Then give that as source.
  • Line 173 and 174 exponents must be clearly marked as such.
  • Chapter 2.2: It would be interesting for the reader what condition at the wall has been used (no slip?)
  • Chapter 2.3: It would be interesting for the reader to know if a rectangular grid had been used , if so then how the grid were varied. (same dimension of cells throughout? Finer cells at wall?,…)
  • Figure 4: The pressure given is 8.5 MPa while in all cases in results it is 8. The Mass flow is 800 kg/(m²*s), while in the cases it is 1500, 2000, 2500. What is the reason? It would be preferable to have the verification at relevant conditions. (Or at conditions as given in Figure 5)
  • Figure 5: The diagrams are doubled…
  • Figure 5: same scale for (a) and (b) is mandatory to maximize clarity
  • Figure 5: indicatin of ipc (as in results-digrams) is recommended
  • Figure 5(b) while the experimental data show a minimum of Tw after ipc, while simulation goes through a maximum. It is advised to comment on that.
  • Chapter2: It is unclear how T+, on which a lot of interpretation relies is calculated. This has to be given otherwise the use of T+ is not to be understood at all! (e.g. is T+-Tb a function of any of the used variables or a constant value)
  • Line 217/218 you report a fact, which can’t be seen from the data. It is immediately clear however. But it might be worthwhile wo show that in a Figure 6(b) [one can derive it by the lengths of the lines, but that is an arduous task, especially since ib is not directly proportional to z)
  • Figure 7: d is defined differently from the use later on (difference between T- and T+ as opposed to difference between wall and T+).
  • Figure 7: If conditions are turbulent flow, why is T still falling in the core flow? It would probably help to show the limit of the laminar boundary layer near the wall.
  • If Tpc would be given it would enlighten the reader if the temperature interval is symmetric to Tpc.
  • Line 246: Tb is shown instead of ib
  • Figure 8: It seems that for the high heat flow the wall temperature reach a constant value after the peak. It might be worthwhile to comment on that.
  • Figure 8/Figure 10: Having the same scaling might add clarity.
  • Line 260: results are presented in Figure 9 not 8
  • Paragraph 257-276. It would increase readability to make several paragraphs one for each point.
  • All Figures 9 and 11. There seems to be a layout problem with the “5” being not in line.
  • Figure 9 (c ) is never discussed. The authors could insert a diagram showing the actual radial temperature profiles. This could be instructive for the reader, especially since the temperature difference between wall and bulk is much more pronounced at high heat flows.
  • Lines 266-268: The effect of l is discussed. However l is mainly relevant when heat conduction is the main mechanism of heat transfer. If convective heat transfer is the main mechanism (as in turbulent conditions) it should not be so relevant. This should be discussed.
  • Lines 272-274: This is the beginning of a discussion about the effect of the laminar undercurrent, and should be elaborated.
  • Figure 10: There seems to be a gap in the Tb-line
  • Lines296-299/Figure11 Probably it would be a plus if we saw the development of density over the radius. That might clear up the “gas-like” and the “liquid-like” property?
  • Figure 11(c ) viscosity is not discussed in the text (same as Figure 9)
  • Equation (12): probably the u should be a n.
  • Figure 12 and 13: There shouldn’t be a zero on the bottom of the ordinate in a linear scale. Put in the correct values please.
  • Conclusions: The conclusions Should be elaborated after the revision of the paper.
  • Nomenclature:the following are missing: A, Bu*, d, ilat, isen, k (a K is given as Greek but if that was the case it should be a K), u. In Greek Symbols: s,d Subscripts: i,j,k,w.
  • Funding and Acknowledgments contain the same text.

Round 2

Reviewer 1 Report

The author did not answer all of my questions accordingly. They just have responded to some parts of issues and have left some others! I give one more chance to authors to answer ALL of my questions/ comments. If they cannot address all of my questions/ comments, I have no choice just to reject the paper.

Reviewer 2 Report

Review2

Kaigang Gong, Bingguo Zhu, Bin Peng, Jixiang He:

Numerical investigation of heat transfer characteristics of scCO2 flowing in a vertically upward tube with high mass flux.

The reviewer thanks the author for the vast amount of work they put in. The paper has improved very much.

  • The use of English language is still weak. Example: 31 “are one new method”…are new methods would be correct.
  • Figure 1: As opposed to what the author wrote, the line for r is still missing
  • Figure 6 (previously Figure 5), the scale of (a) and (b) are still different
  • Several symbols are still missing in Nomenclature: A, Bu*, d and k as well as s and d
